# Chromatin-Directed Proteomics Identifies ZNF84 as a p53-Independent Regulator of p21 in Genotoxic Stress Response

**DOI:** 10.3390/cancers13092115

**Published:** 2021-04-27

**Authors:** Anna Strzeszewska-Potyrała, Karolina Staniak, Joanna Czarnecka-Herok, Mahmoud-Reza Rafiee, Marcin Herok, Grażyna Mosieniak, Jeroen Krijgsveld, Ewa Sikora

**Affiliations:** 1Laboratory of Molecular Bases of Aging, Nencki Institute of Experimental Biology, PAS, 3 Pasteur Street, 02-093 Warsaw, Poland; k.kucharewicz@nencki.edu.pl (K.S.); j.czarnecka@nencki.edu.pl (J.C.-H.); marcin.herok@gmail.com (M.H.); g.mosieniak@nencki.edu.pl (G.M.); e.sikora@nencki.edu.pl (E.S.); 2Bioinformatics and Computational Biology Laboratory, Francis Crick Institute, 1 Midland Road, London NW1 1AT, UK; mahmoud-reza.rafiee@crick.ac.uk; 3German Cancer Research Center (DKFZ), Division of Proteomics of Stem Cells and Cancer, Im Neuenheimer Feld 581, 69120 Heidelberg, Germany; j.krijgsveld@dkfz-heidelberg.de; 4Medical Faculty, Heidelberg University, Im Neuenheimer Feld 672, 69120 Heidelberg, Germany

**Keywords:** ZNF84, p21, *CDKN1A*, genotoxic stress, senescence, chemotherapy, chromatin pulldown, mass spectrometry

## Abstract

**Simple Summary:**

Chemotherapy is a commonly applied anticancer treatment, however therapy-induced senescent growth arrest has been associated with aggressive disease recurrence. The p21 protein, encoded by *CDKN1A*, plays a vital role in the induction of senescence. Its transcriptional control by p53 is well-established. However, in many cancers where *TP53* is mutated, p21 expression must be triggered by p53-independent mechanisms. We here used a chromatin-directed proteomic approach and identified ZNF84 as a regulator of *CDKN1A* gene expression in various p53-deficient cell lines. Knock-down of ZNF84, an as-yet un-characterized protein, inhibited p21 gene and protein expression in response to doxorubicin and facilitated senescence bypass. Intriguingly, ZNF84 depletion diminished genotoxic burden evoked by doxorubicin. Clinical data association studies indicated the relevance of ZNF84 expression for patient survival. Collectively, we identified ZNF84 as a critical regulator of senescence-proliferation outcome of chemotherapy, opening possibilities for its targeting in novel anti-cancer therapies of p53-mutated tumours.

**Abstract:**

The p21^WAF1/Cip1^ protein, encoded by *CDKN1A*, plays a vital role in senescence, and its transcriptional control by the tumour suppressor p53 is well-established. However, p21 can also be regulated in a p53-independent manner, by mechanisms that still remain less understood. We aimed to expand the knowledge about p53-independent senescence by looking for novel players involved in *CDKN1A* regulation. We used a chromatin-directed proteomic approach and identified ZNF84 as a novel regulator of p21 in various p53-deficient cell lines treated with cytostatic dose of doxorubicin. Knock-down of ZNF84, an as-yet un-characterized protein, inhibited p21 gene and protein expression in response to doxorubicin, it attenuated senescence and was associated with enhanced proliferation, indicating that ZNF84-deficiency can favor senescence bypass. ZNF84 deficiency was also associated with transcriptomic changes in genes governing various cancer-relevant processes e.g., mitosis. In cells with ZNF84 knock-down we discovered significantly lower level of H2AX Ser139 phosphorylation (γH2AX), which is triggered by DNA double strand breaks. Intriguingly, we observed a reverse correlation between the level of ZNF84 expression and survival rate of colon cancer patients. In conclusion, ZNF84, whose function was previously not recognized, was identified here as a critical p53-independent regulator of senescence, opening possibilities for its targeting in novel therapies of p53-null cancers.

## 1. Introduction

Multiple genotoxic agents can induce a senescent phenotype, a state of persistent proliferation-arrest characterized by e.g., cell enlargement and increased activity of senescence-associated β-galactosidase (SA-β-gal) [1] (reviewed in [2]). Senescence occurring in response to genotoxic chemotherapeutics or radiotherapy is known as therapy-induced senescence (TIS), which has not only been observed in *in vitro* culture, but also *in vivo* in patients after anticancer therapy, e.g., in breast tumours after cyclophosphamide, doxorubicin, and 5-fluorouracil treatment [3], and in non-small cell lung cancer patients who received neoadjuvant chemotherapy carboplatin and taxol prior to surgery [2,4]. Because senescent cells are growth-arrested, induction of senescence in cancer cells has been perceived as a potentially desirable response to therapy, albeit being solely tumour-suppressive and not curative (reviewed in [5]). However, recent evidence challenged this belief, since senescent cells are apoptosis-resistant (as a direct consequence of the sustained up-regulation of p21^WAF1/Cip1^, named p21 in the rest of this manuscript) [6,7], and their senescence-associated secretory phenotype (SASP) often confers metastasis-promoting properties [8]. SASP contributes also to chronic inflammation, leading to alterations in the tissue micro-environment and to tissue impairment [9,10], thus rendering a risk for the organism. On top of that, the previous paradigm about irreversibility of senescence itself was recently questioned. Multiple studies reported resumption of proliferation by senescent cells, concomitant with enhanced aggressiveness [2]. It has been argued that TIS of cancer cells could be an alternative form of dormancy, allowing cancer cells to survive stress evoked by anticancer-therapy, thus creating a barrier to treatment efficacy [5], and ultimately leading to re-emergence of tumours with more aggressive properties. As cellular senescence is such a multi-dimensional, apparently health-threatening and relatively common phenomenon, it is important to determine its molecular basis and regulatory mechanisms. 

The most commonly used inducers of senescence are genotoxic agents causing DNA double strand breaks (DSBs), and, concordantly, permanent activation of DNA damage response (DDR) is considered to be a trigger and a hallmark of cell senescence [11]. The p21 protein, an inhibitor of cyclin-dependent kinases, is the effector of the canonical (ATM-p53-p21) DDR axis [12,13]. Its elevated expression has been causally associated with the induction of senescence, both in normal [14] and in cancer cells [15]⁠. In addition, senescence has been functionally linked to aging (reviewed in [16]) and, concordantly, elevated p21 protein expression was observed e.g., in liver and kidney of normally ageing mice [17] and in fibroblasts from patients suffering from premature aging syndrome [18]. Conversely, loss of p21 counteracts senescence both in cultured cells [19] and *in vivo* [20], and deletion of the *CDKN1A* gene had anti-aging effects in telomerase-deficient mice, it improved stem cell function and prolonged lifespan [21]. Collectively this emphasizes the central role of p21 in senescence (and aging), mediated by its ability to arrest cell-cycle.

The main transcriptional regulator of p21 in senescence is p53 (encoded by *TP53*), which is phosphorylated on Ser15 in response to DNA damage. Accordingly, p21 is considered to be the canonical effector of p53 activation [22]. Nevertheless, it has been shown by many groups, including ours, that trans-activation of *CDKN1A* can occur in cells in which p53 is absent or malfunctioning, leading to p53-independent senescence [1,23,24]. Interestingly, p21’s activity in a p53-deficient background is emerging as particularly damaging, by triggering and fuelling the path to aggressive cancer phenotype [25]. Accumulated evidence shows that p21 production is activated by a wide range of p53-independent signals and stimuli, including growth factors, nuclear receptors, chemicals and drugs [26,27], yet, according to our unpublished studies aimed at explaining the mechanism of p21 elevation in senescent p53-deficient HCT116 cells, this picture of p21 regulation was incomplete. 

Here, we aimed to characterize novel molecular mechanisms enabling p53-independent activation of the *CDKN1A* gene in cancer cells induced to undergo senescence. In particular, we applied a mass spectrometry-based approach to identify chromatin-bound proteins occupying the promoter region of *CDKN1A*, aiming to identify novel factors that regulate *CDKN1A* expression. This identified ZNF84, an as-yet uncharacterized protein containing zinc-finger motifs, characteristic of transcription factors. In order to elucidate the role of this protein, functional characterization was performed, and it confirmed that ZNF84 controls *CDKN1A* expression and cell senescence, in a manner that is independent of p53. This study resulted in finding novel mechanism driving p53-deficient cells into growth arrest state which diminishes the effectiveness of chemotherapy and characterized for the first time the function of ZNF84 protein.

## 2. Materials and Methods

### 2.1. Cell Culture and Treatments

The human HCT116 colon cancer cell lines (p53WT and p53KO) were kindly provided by Prof. Bert Vogelstein (Johns Hopkins University, Baltimore, MD, USA). The lung cancer cell lines (H358 [CRL-5807] and H1299 [CRL-5803]), as well as breast cancer cell line MCF7 [HTB-22] were obtained from ATCC. Authentication of cell lines was performed by Cell Line Authentication IdentiCell STR using STR profiling. Cells were grown under standard conditions (37 °C, 5% CO2) in McCoy’s (HCT116), RPMI 1640 (H358, H1299) or Dulbecco’s modified Eagle’s low glucose medium (MCF7), supplemented with 10% FBS and antibiotics. For transfection cells were seeded at a density 5 × 10^3^/cm^2^ (HCT116, MCF7) or 4 × 10^3^/cm^2^ (H358 and H1299) on 6-well plates. After 24 h they were transfected with siRNA. Two different anti-ZNF84 siRNA sequences were tested, namely #s15134 and #s15132 (Ambion, Thermo Fisher Scientific, Waltham, MA, USA), marked as siZNF and siZNF_II, respectively. As no significant differences between those two sequences were observed (similar efficiency in gene expression silencing (Appendix A), as well as in lowering the level of p21 and ZNF84 proteins and downregulation of senescence marker SA-β-gal (data not shown)), for more extended analysis siZNF was chosen. Cells were transfected with 30 nM siRNA (siZNF or non-targeting negative control siRNA, siNeg) (Ambion, Thermo Fisher Scientific) using Lipofectamine 2000 (Thermo Fisher Scientific), according to the manufacturer’s protocol. 24 h after transfection medium was replaced with fresh one and after the next 24 h doxorubicin was added (Sigma Aldrich, Saint Louis, MO, USA). The doses of doxorubicin which induce premature senescence were set as described by us previously [24,28]—the final concentration of doxorubicin was 100 nM (in case of HCT116, H1299, MCF7) or 50 nM (H358 cells). Cells were cultured in the presence of the drug for up to 5 days. 

### 2.2. DNA Content and Cell Cycle Analysis

For DNA analysis the cells were fixed in 70% ethanol and stained with PI solution (3.8 mM sodium citrate, 500 µg/mL RNAse A, 50 µg/mL PI in PBS). All of the agents were purchased from Sigma Aldrich. DNA content was assessed using flow cytometry (FACS Calibur, Beckton Dickinson, Franklin Lakes, NJ, USA). 10,000 events were collected per sample. Data was analyzed with CellQuest Software (Beckton Dickinson) and ModFit Software (Verity Software House, Topsham, ME, USA).

### 2.3. Western Blotting Analysis

Whole cell protein extracts were prepared using the SLB buffer (50 mM Tris-HCl pH 6.8, 10% glycerol, 2% SDS). The amount of protein was measured using bicinchoninic acid (BCA) protein assay kit (Sigma Aldrich) according to the manufacturer’s instructions. Afterwards, DTT and bromophenol blue were added (final concentration was 52 mM and 0.01%, respectively) and samples were denatured by incubation in 95 °C (10 min). Equal amounts of protein (20 µg) were separated electrophorectically in 12% SDS-polyacrylamide gel and afterwards transferred to nitrocellulose membrane. Membrane was blocked for 1 h at RT either in 5% BSA (Lab Empire, Rzeszow, Poland) or 5% non-fat milk dissolved in TBS containing 0,1% Tween-20, depending on the primary antibody used in the next step. Anti-ZNF84 antibody (Thermo Fisher Scientific) required blocking a membrane in BSA solution and for all other antibodies a membrane was blocked in milk. The following antibodies were used: anti-ZNF84 (1:500) (Thermo Fisher Scientific), anti-p53 (DO-1) (1:500) (Santa Cruz Biotechnology Inc., Dallas, TX, USA), anti-p21 (C-19) (1:500) (Santa Cruz Biotechnology Inc.) and anti-GAPDH (1:50.000) (Merck-Millipore, Burlington, MA, USA). The proteins were detected with appropriate secondary antibodies conjugated with horseradish peroxidase and ECL reagents (Thermo Fisher Scientific), according to the manufacturer’s protocol.

### 2.4. Cytochemical Detection of SA-β-Gal

Detection of SA-β-Gal was performed according to Dimri et al. [29]. Briefly, cells were fixed with 2% formaldehyde, 0.2% glutaraldehyde in PBS, washed and exposed overnight at 37 °C to a solution containing: 1 mg/mL 5-bromo-4-chloro-3-indolyl-b-D-galactopyranoside, 5 mM potassium ferrocyanide, 5 mM potassium ferricyanide, 150 mM NaCl, 2 mM MgCl2 and 0.02 M phosphate buffer, pH 6.0. All of the chemicals were purchased from Sigma Aldrich. Photos were taken in transmitted light using the Nikon Eclipse Ti-U fluorescent microscope and Nikon Digital Sight DS-U3 camera (Nikon, Tokyo, Japan).

### 2.5. Flow Cytometric Detection of SA-β-Gal 

SA-β-gal activity was measured by flow cytometry, using the fluorogenic substrate 5-dodecanoylaminofluorescein di-β-D-galactopyranoside (C12FDG) according to Kurz et al. [30]. Briefly, 100.000 cells were incubated with 33 μM C12FDG (LifeTechnologies, Thermo Fisher Scientific) in 37 °C for 30 min. Afterwards cells were washed with PBS and analyzed immediately with BD FACS Calibur flow cytometer (Beckton Dickinson). Data was analyzed using the CellQuestPro software. SA-β-gal activity was estimated using the geometric mean fluorescence intensity of the population.

### 2.6. Quantitative Real-Time PCR (RT-PCR)

Real time quantitative-PCR was used to quantify the gene expression of mRNAs for *CDKN1A* and *ZNF84* (with expression of *GAPDH* used as endogenous control). Total RNA was isolated from cells with RNeasy Micro Kit (Qiagen, Wroclaw, Poland) according to the manufacturer’s instruction. First-strand cDNA was synthesized using 0.5 μg of total RNA. cDNA was mixed with SYBR Green Master Mix (Thermo Fisher Scientific) and 1 μM starters for: human *CDKN1A* (forward: 5’-AAGACCATGTGGACCTGTCACTGT-3’, reverse: 5’-GAAGATCAGCCGGCGTTTG-3’), human *ZNF84* (forward: 5’-AGAACCGATCTCAGCCTTGC-3’, reverse: 5’-CATAACTTCATACCCCAGTGACA-3’) and human *GAPDH* (forward: 5’-TGCACCACCAACTGCTTAGC-3’, reverse: 5’-GAGGGGCCATCCACAGTCTTC-3’). The reactions were performed with the use of StepOnePlus™ Real-Time PCR System (Thermo Fisher Scientific). Results were analyzed using relative quantification—the ΔΔCt approximation method.

### 2.7. Luciferase Reporter Assay

HCT116 p53KO cells were seeded in 6-well plates (10.000 cells per cm^2^) and transfected with 2 μg of WWP‑Luc (p21/WAF1 promoter) vector encoding firefly luciferase reporter fused to the p21 promoter. The plasmid was a gift from Prof. Bert Vogelstein ([22]; Addgene plasmid # 16451). Fugene (Promega, Madison, WI, USA) was used as a transfection reagent, according to the manufacturer’s protocol. After 2 days of transfection, the cells were reseeded to 12-well plates (5000 cells per cm^2^) for siRNA transfection, which was performed as described previously. A fraction of wells was treated with 100 nM doxorubicin for 2 days before luminescence measurement, others were left untreated. Lysates were analyzed by a luciferase reporter assay system (Promega) according to the manufacturer’s protocol, in technical triplicate, using GloMax 20/20 luminometer (Promega). The results were normalized to the total protein level, assessed by the BCA assay.

### 2.8. ELISA Assay

To assess the secretion of IL-8 proteins culture medium was collected and subjected to analysis according to the manufacturer’s instructions (R&D Systems, Minneapolis, MN, USA). Absorbance was measured at 450 nm with the use of a Tecan Sunrise spectrophotometer with X-fluor software (Tecan, Männedorf, Switzerland).

### 2.9. BrdU Incorporation

Cells, seeded on cover glasses in 12-well plates, were treated with 10µM BrdU for the last 18 h of culture. Next, they were washed with PBS and fixated in ice-cold 70% ethanol. Cells were washed with 0.5% TritonX-100 (Sigma-Aldrich) in PBS, incubated in 2N HCl for 30 min, washed twice with PBS, incubated for 1 min in 0.1 M borax solution (Sigma-Aldrich), washed twice in PBS again and incubated with primary anti-BrdU antibody (Beckton Dickinson) 1:120 in 1% BSA 0.5% Tween-20 solution in PBS for 1 h. Then, cells were washed twice with 0.5% Tween-20 in PBS and incubated with secondary antibody conjugated to fluorochrome (Thermo Fisher Scientific). Cells were washed with 0.5% Tween-20 in PBS, stained with 1 M DAPI for 15 min, washed again and mounted on a microscope slide with FluoroMount Medium (Thermo Fisher Scientific). Specimens were examined under an Eclipse fluorescence microscope (Nikon, Tokyo, Japan). Images were analyzed using ImageJ program. Number of BrdU-positive cells was calculated relative to the number of all cells (based on DAPI staining).

### 2.10. Nuclear Area

Nuclear area was calculated on the basis of the images of DAPI staining, taken with Nikon Eclipse fluorescent microscope and Nikon Digital Sight DS-U3 camera (Nikon, Tokyo, Japan). Analysis was performed with the use of ImageJ software.

### 2.11. γH2AX Immunofluorescent Staining

Before fixation of cells growing on cover-slips, cell culture medium was replaced with serum-free medium containing 2.5µM CellTracker™ Red CMTPX fluorescent dye (Thermo Fisher Scientific). Staining was performed according to the manufacturer’s protocol. Afterwards, cells were washed with PBS and fixed in 4% paraformaldehyde for 10 min. Cell membranes were permeabilised by 10 min incubation in 0.5% Triton X-100 in PBS, then cells were blocked in 2% BSA, 1.5% goat serum, 0.1% Triton X-100 in PBS for 10 min. Afterwards, cells were incubated on slides with an anti-γH2AX antibody (Abcam, Cambridge, UK) 1:500 in blocking buffer (2% BSA, 1.5% goat serum, 0.1% Triton X-100 in PBS). Next, secondary Alexa 488-conjugated IgG antibody (1:500) was used (Life Technologies, Carlsbad, CA, USA). To visualize the nuclei, DNA was stained using Hoechst (2 μg/mL in PBS) (Life Technologies). The specimens were analyzed using Zeiss Spinning Disc confocal microscope (Zeiss, Oberkochen, Germany) using a 63×/1.4 oil immersion lens. Fluorescence was excited using the 405-nm line from a pulsed laser (for Hoechst), 488 laser (for Alexa 488) and 561 laser (for CellTracker™). Confocal z section stacks were collected at 0.22-μm spacing through the depth of the specimen. For unbiased estimation of the number of γH2AX foci, images ware analyzed in an automated manner, in each z layer separately, filtering out blobs >45 μm^2^ and nuclei with homogeneously diffused signal. Analysis was performed with the use of ImageJ software.

### 2.12. Colony Formation Assay

Cells were seeded, transfected and treated with doxorubicin as described in the chapter Cell culture and treatments, but instead of constant 5-day culture in presence of doxorubicin, after 24 h the doxorubicin-containing medium was replaced with drug-free one and cells were cultured for 7 days. Afterwards, cells were replated at density low enough to allow for the formation of single colonies within the next 7 days (330 live cells per a single well of a 6-well plate, in technical triplicate). The colonies were fixed with methanol, stained with 4% methylene blue and counted.

### 2.13. MPM-2

Cells, seeded and transfected as described in the chapter Cell culture and treatments, were collected after 24 h of doxorubicin treatment. 4 h before the collection of cells, nocodazole (200 ng/mL) was added in order to facilitate accumulation of mitotic cells. Detached by trypsinization, cells were fixed in 4% PFA solution in PBS and stored in 70% ethanol in −20 °C until further analysis. Then, cells were washed, incubated in blocking buffer (3% BSA, 0.1% Triton X-100 in PBS) for 1 h and stained for 2 h with an anti-phospho-Ser/Thr-Pro MPM-2 antibody (1μg/mL) (Merck-Millipore, Burlington, MA, USA) dissolved in 1% BSA, 0.1% Triton X-100 in PBS. As a secondary antibody, Alexa Fluor^®^ 488 goat anti-mouse IgG (4 μg/mL) (Thermo Fisher Scientific) was used. Intensity of cell fluorescence in FL1 channel was measured by flow cytometry (FACS Calibur with CellQuest Software—Beckton Dickinson). At least 10 × 10^3^ cells were analyzed in each sample.

### 2.14. RNA Isolation and Sequencing

RNA was isolated using RNeasy^®^ Plus Mini Kit (Qiagen, Venlo, The Netherlands). RNA integrity was checked using the RNA Nano 6000 Assay Kit of the Bioanalyzer 2100 system (Agilent Technologies, Santa Clara, CA, USA), and concentration was measured with Qubit^®^ RNA Assay Kit in Qubit^®^ 2.0 Fluorometer (Life Technologies). Stranded mRNA-Seq libraries were prepared from 300 ng of total RNA using the Illumina TruSeq RNA Sample Preparation v2 Kit (Illumina, San Diego, CA, USA) implemented on the liquid handling robot Beckman FXP2. Obtained libraries that passed the QC step, which was assessed on the Agilent Bioanalyzer system, were pooled in equimolar amounts. 1.8 pM solution of each pool of libraries was loaded on the Illumina sequencer NextSeq 500 High output and sequenced uni-directionally, generating ~450 million reads per run, each 85 bases long. All samples were aligned to the GRCh38 genome using the STAR RNA-seq aligner. Library preparation, RNA sequencing and alignment were performed by the EMBL Genomics Core Facility (Heidelberg, Germany).

### 2.15. Bioinformatic Analysis of Transcriptomic Data

Aligned gene counts were analyzed using the edgeR package [31] (Bioconductor release version 3.10) in R software environment (version 3.6.2). The cpm function from the edgeR was used to normalize for the different sequencing depths for each sample. TMT (Trimmed Mean of M values) normalization was performed to eliminate composition biases between libraries [32]. The voom function from the limma package (version 3.44.3) was used to transform the read counts into logCPMs (counts-per-million) while taking into account the mean-variance relationship in the data [33]. Then, voom transformed data were fitted into linear model to test for differentially expressed genes. Gene set enrichment analysis (GSEA) [34] was performed using fgsea algorithm [35]. Input for the analysis was list of all the genes which passed quality control, sorted from most significantly up-regulated to most significantly down-regulated genes and the input annotation of Term-To-Gene mapping was downloaded in version v7.0 from https://www.gsea-msigdb.org (accessed on 8 December 2019), provided by the Broad Institute and UC San Diego (H collection of hallmark gene sets, C2 collection of curated gene sets—CP KEGG and REACTOME databases, as well as selected C2 sub-collections CGP: Chemical and Genetic Perturbations, as indicated in the figure descriptions). Normalized enrichment scores (NES) with FDR-adjusted *p*-values < 0.05 were considered statistically significant. Plots were prepared using plotEnrichment function from fgsea package. Pathview package [36] was used to render gene expression data on relevant KEGG database pathway graphs, by converting it to pseudocolors on a scale from the most repressed (green, −1 in arbitrary units) to most up-regulated (red, +1 in arbitrary units). Other visualizations were performed with ggplot2 and pheatmap packages.

### 2.16. Proteomics of Isolated Chromatin Fragments (PICh)

HCT116 p53KO cells were seeded at density of 10 × 10^3^/cm^2^. 24 h later, senescence-inducing dose of doxorubicin (100 nM) was added and cells were cultured in presence of the drug for 2 days. PICh protocol with modifications was carried out based on Rafiee MR doctoral dissertation (Chapter 5). Briefly, cells were detached from culture dishes by rubber policeman and DNA-protein complexes were cross-linked by incubating cells in 1.5% formaldehyde in PBS for 14 min. Cross-linking was stopped by adding glycine solution to the final concentration of 120 mM and incubating for 5 min, then cells were spinned down, washed and counted. In total, 48 × 10^6^ cells were used per one experiment, of which 24 × 10^6^ cells were used for *CDKN1A* gene promoter isolation and the remaining 24 × 10^6^ for negative control. Cell pellet was suspended in IP buffer (50 mM Tris-HCl pH 8.0, 5 mM EDTA, 1% Triton, 0.5% NP40) with protease inhibitors (Roche, Basel, Switzerland) and incubated for 15 min in 37 °C with RNase A (Thermo Fisher Scientific) with gentle agitation (400 rpm). After spinning down (2 min, 2000× *g*), pellet was suspended in DNAzol (Thermo Fisher Scientific), briefly vortexed and centrifuged (2 min, 2000× *g*, 4 °C). Then DNAzol treatment procedure was repeated. The pellet, containing DNA with cross-linked proteins, was suspended in freshly-prepared 25 mM NaOH and incubated in 37 °C for 30 min to facilitate denaturation. Then, the sample was sonicated using Bioruptor Pico device in dedicated tubes (Diagenode, Seraing, Belgium) (5 cycles of 30 s ON and 30 s OFF, high mode) in order to obtain DNA fragments of mostly ~200–500 bp. Each sample (corresponding to 24 million cells) was incubated with 6 pmol of either *CDKN1A* targeting (“F2”, GTAAACCTTAGCCTGTTACTCTGAACAGGGTATGTGATCTGCCAGCAGAT) or reversed, negative control probe (TAGACGACCGTCTAGTGTATGGGACAAGTCT CATTGTCCGATTCCAAATG), denatured prior to use by incubation in 95 °C and immediate cooling on ice. Then mixtures of samples with probes were transferred to 3k Amicon ultrafilteration columns (Merck Millipore) and spun at 12,000× *g* in 4 °C for 2.5 min to concentrate the sample. Contents of the Amicon inserts were then transferred to 30 k Amicon columns and spun at 12,000× *g* in 4 °C for 5 min. One volume of BW buffer (10 mM TrisCl pH 8.0, 1 mM EDTA, 0.1% Triton ×100, 1 M NaCl) was added to each insert to promote renaturation and spun again for 8 min in the same conditions. Another volume of BW buffer was added, samples were transferred to tubes, incubated for 5 min in 70 °C, then left it in RT for 15 min to slowly cool down. Samples were transferred to 15 mL tubes, mixed with BW buffer to the final volume of 10 mL and 100 µL of streptavidin beads (New England Biolabs, Ipswich, MA, USA; pre-prepared by blocking for the MS experiment, as described in [37]) was added to each of the tubes. Samples were incubated in RT overnight on a rotator wheel. On the next day the washes were performed using magnetic stand: twice in 7 mL of SDS Wash buff 1 (10 mM TrisCl, 1 mM EDTA, 0.5% SDS, 500 mM NaCl), once in 5 mL of SDS Wash buff 2 (10 mM TrisCl, 1 mM EDTA, 1% SDS, 200 mM NaCl) and twice in 7 mL of Isopropanol wash buffer (20% 2-propanol in water, 500 mM NaCl). The beads were re-suspended in 1.5 mL of Isopropanol wash buffer, transferred to 2 mL tube, separated on the magnetic stand, re-suspended again in isopropanol wash buffer (80 µL) and transferred to PCR tubes. The liquid was discarded and the beads re-suspended in 13 µL digestion buffer (10 mM TrisCl, 1 mM EDTA, 0.1% SDS). In the pilot experiment at this point de-crosslinking was performed in order to evaluate the efficiency of isolation of the DNA fragment around *CDKN1A* TSS by RT-PCR. To this end samples were incubated in 65 °C overnight. For RT-PCR the following primers were used: CDKN1A_F CTTTCTGGCCGTCAGGAACA, CDKN1A_R ATCAGATCCCAGCCCTGTCGCA and control primers targeting unrelated *FOSL1* gene (FOSL1_F TCCCACATCCAACTCCAGCAAC, FOSL1_R AAGCTGGCTCTACTGTGAAGCAC). As a result of RT-PCR analysis fold of enrichment in TIGER reaction with F2 vs. negative control probe was calculated to be 39.62.

In the full experiment, after re-suspending samples in digestion buffer, they were prepared for mass spectrometric analysis. To this end 1 μL of 100 mM DTT was added to the samples, which then were incubated at 95 °C for 20 min. Afterwards iodoacetamide was added (1 μL of 200 mM IAA) and samples were incubated at room temperature for 30 min. After addition of another 1μL of 100 mM DTT, beads were removed and cells were incubated for 14 h at 37 °C with MS-grade trypsin (Promega, Madison, WI, USA) (3 μL). Afterwards the samples were cleaned with the SP3 protocol [38] prior to proteomic analysis. Single-run proteome analysis was performed on a Qexactive HF mass spectrometer (Thermo Scientific). For peptide identification Max Quant software was used and the proteinGroups.txt table was filtered to remove entries in ‘potential contaminants’ and ‘reverse’. ZNF84 was identified by 1 unique peptide in MS/MS spectra for F2 sample, absent in the negative control sample. Q value was 0.0072.

### 2.17. Colon Adenocarcinoma Patients’ Survival Analysis

Colon adenocarcinoma (COAD) patients clinical and genetic data were retrieved from The Cancer Genome Atlas (TCGA) using the UCSC Xena Browser (http://xena.ucsc.edu; accessed on 17 August 2020) [39]. After proper filtering (Study: TCGA Colon Cancer (COAD); Phenotypic: sample_type—Primary Tumor, OS.time, OS; Genomic: Gene Expression–ZNF84) data of 283 patients were used for survival analysis. Patients were divided into two groups (ZNF84 high (*n* = 143) and low (*n* = 140)) by median value of ZNF84 gene expression. Significant difference in survival was assessed by log-rank test, final Kaplan Meier plot was prepared using GraphPad Prism software (v. 8.3.1., GraphPad Software, San Diego, CA, USA).

### 2.18. Statistical Analysis

Data distribution normality was analyzed with Shapiro-Wilk test and statistical significance was analyzed with the Student’s *t*-test. Details of individual analysis are stated in the figures’ descriptions. Statistical significance is indicated in the pictures with asterisks (*p*-value  <  0.05 indicated with single * mark, *p*-value  <  0.01—double mark, *p*-value < 0.001—triple mark *p*-value  <  0.0001—quadruple mark). 

## 3. Results

To explore which proteins occupy the DNA regulatory region of *CDKN1A* gene in p53 deficient colon cancer HCT116 (p53KO) cells, we used a proteomic approach enabling the identification of proteins that associate with a genomic region of interest [40,41,42]. Specifically, we designed a biotinylated probe to bind to a region in close proximity (~100 nucleotides downstream) of the *CDKN1A* transcription start site (TSS) (Figure 1A). With this probe we aimed to capture chromatin fragments located within ~1 kb around *CDKN1A* TSS, as a direct result of the size of chromatin fragments generated by sonication (Figure 1A). As a negative control, we used a probe with reversed sequence of the targeting probe, which does not have homology to any region in the human genome. Since we were particularly interested in proteins that bind the locus upon senescence induction, we performed the experiment in cells that had been treated with doxorubicin for 2 days, a time point at which p21 protein level builds up before reaching its maximum on day 3 [24]. 

When examining the results of mass spectrometry, we focused on those proteins for which peptide intensities were increased in the sample incubated with the *CDKN1A*-specific (“F2”) probe as compared to the reversed probe. ZNF84 was among the proteins identified exclusively with the F2 probe, together with nine other proteins containing zinc-finger motifs (Figure 1B). Zinc-finger proteins (ZNFs) are often implicated in transcriptional regulation, among an increasing number of other functions [43]. Since ZNF84 was considered a putative transcriptional regulator [44], however with a poorly characterized function, we focused on exploring its role in *CDKN1A* expression and cell senescence.

First, we aimed to verify whether ZNF84 is involved in the regulation of *CDKN1A* gene expression in cells undergoing chemotherapy-induced senescence. To this end we employed RNA interference to silence ZNF84 gene expression in HCT116 p53KO cells (two siRNA sequences were tested and both had comparable levels of knock-down, Appendix A). After 48 h upon siRNA transfection we treated cells with doxorubicin (Dox) for 2 days, followed by RT-PCR on total RNA (Figure 1C).

When comparing cells transfected with siRNA silencing ZNF84 (siZNF) vs. negative control siRNA (siNeg), we observed a more than 50% decrease in the level of ZNF84 transcript in siZNF cells and similarly decreased level of *CDKN1A* (Figure 1C). This indicates that *CDKN1A* expression depends on ZNF84, either directly or indirectly. Similar conclusion can be drawn from the analysis of *CDKN1A* promoter activity by luciferase reporter assay (Appendix A)

Next, we aimed to explore whether this decrease is detectable also on the protein level. Therefore we collected cell pellets and performed immunoblotting with antibody against p21 protein (Figure 1D). In Dox-treated HCT116 p53KO cells with down-regulated ZNF84 the level of p21 was undetectable. We extended this analysis by including additional p53-deficient cell lines (H358 and H1299, both non-small cell lung cancer cell lines; p53 analysis in Appendix A) as well as cells possessing wild-type p53 protein (HCT116 p53WT and breast cancer MCF7 cell line). ZNF84 silencing consistently attenuated p21 expression, and was most evident in p53-deficient cells, whereas the presence of wild-type p53 diminished this response to various extent, depending on the cell line: while in HCT116 p53WT mild decrease of p21 was observed, MCF-7 failed to down-regulate the level of this protein. This deviating effect in MCF7 may be a cell-type specific effect of the presence of p53, which plays a major role in *CDKN1A* trans-activation and/or could be explained by the possible involvement of breast cancer oncogenes PRMT6 [45] and FHL2 [46], as well as IGFBP-rP1 [47], which comprise p53-independent mechanisms of p21 control.

In summary, our results indicate that ZNF84 is involved in genotoxic stress response via regulation of p21 expression, in cancer cells originating from different tissues and with various genetic backgrounds. The observation that this effect was less pronounced in p53-proficient cells indicates that ZNF84-dependent expression of p21 is activated as an alternative pathway when p53 is lost.

### 3.1. ZNF84-Deficient Cells Exhibit Attenuation of Therapy-Induced Senescence

Considering the crucial role of p21 for cellular senescence, we analyzed the impact of ZNF84 silencing on several markers of senescence. As the senescent phenotype takes several days to fully develop, we cultured HCT116 p53KO cells for 5 days under constant doxorubicin (according to previously established protocol described in [24]) and analyzed markers of senescence starting from day 3. 

Increased activity of senescence-associated-β-galactosidase (SA-β-Gal) remains the fundamental marker of senescence, despite known limitations, such as false-positive results in over-confluent cell cultures [48]. Flow-cytometric analysis indicated that the mean value of SA-β-Gal activity was lower in siZNF cells than in siNeg cells (Figure 2A). This indicates that ZNF84 deficiency may attenuate senescence, possibly by preventing p21 induction (Figure 1C,D), which we aimed to corroborate in a next set of experiments. 

Senescent cells harbor DNA lesions, which persist and cannot be repaired [49]. To assess the occurrence of DNA double strand breaks (DSBs) and repair foci, we probed cells for γH2AX by immunofluorescent detection. In siZNF cells treated with Dox we observed a significantly reduced mean number of γH2AX foci per nucleus, as compared to siNeg cells treated in the same way (Figure 2B and Appendix A).

Similarly, the mean intensity of γH2AX signal per nucleus area was reduced in siZNF cells (Figure 2B), pointing to reduced DNA damage signaling which may (but does not have to) be a result of decreased DNA damage levels in these cells. In addition, Dox-treatment of siZNF cells led to less pronounced nuclear enlargement compared to siNeg controls, which exhibited cell flattening combined with nuclear swelling, as is typical for senescence (Figure 2C). These data strongly indicate that the development of senescent phenotype in response to Dox is reduced due to ZNF84 deficiency. 

Senescent cells cease DNA replication, which can be observed by attenuation of BrdU incorporation into DNA. While the percentage of BrdU-positive cells among Dox-treated siNeg HCT116 p53KO was 16%, the number has more than doubled (38%) in their siZNF counterparts (Figure 2D), again indicating a vital role of ZNF84 in senescence. 

Senescence-associated secretory phenotype (SASP) is another hallmark accompanying senescence, recognized for its clinical implications (e.g., promotion of metastasis). Contrary to the previously described markers, SASP is a complex phenomenon, with diverse composition (set of secreted factors) and various absolute level of secretion per cell, depending on e.g., cell type [50]. Importantly, SASP, although functionally linked to senescence, is regulated in a manner independent of p21 and of senescent cell cycle arrest [51], through DDR-dependent and independent mechanisms [52]. Based on our previous work on HCT116 cells [24] we monitored the pro-inflammatory cytokine IL-8 to assess the impact of ZNF84 down-regulation on SASP. In HCT116 p53KO cells treated with Dox, IL-8 levels were the same irrespective of the siRNA used for transfection (Appendix A), suggesting that ZNF84 has no role in SASP. This result was replicated in another p53-deficient cell line, H358 (Appendix A). Also, transcriptomic analysis of 24 common SASP genes indicated no significant change in expression of most of them in siZNF HCT116 p53KO cells, and even an increase of 2 SASP gene transcripts (MMP1 and TGFB1) (Appendix A). Nevertheless, it cannot be excluded that ZNF84 regulates a subset of SASP factors beyond those which were analyzed.

Collectively these data indicate that ZNF84 is necessary for the development of senescent phenotype, as determined by various hallmarks of this process (SA-β-Gal, DSBs signaling, characteristic cell morphology, cessation of DNA synthesis), but not necessarily SASP.

### 3.2. ZNF84 Deficiency Alters the Expression of Genes Related to Stress Response and Cell Growth

We next employed transcriptome analysis of HCT116 p53KO cells to gain more insight into the molecular processes occurring upon ZNF84 down-regulation followed by Dox treatment. PCA analysis showed a clear segregation of the samples by siRNA used for transfection (siZNF or siNeg) (Figure 3A), as expected. The level of ZNF84 transcript was reduced to ~50% (log2 of change = −0.920, FDR-controlled *p*-value = 0.018), similar to *CDKN1A* (log2 of change = −0.917, FDR-controlled *p* value = 0.0009) (Figure 3B), mirroring our earlier observation by RT-PCR (Figure 1C). Using a FDR-adjusted *p*-value < 0.05 we observed 3374 genes that were differentially expressed between siZNF and siNeg transfected cells (out of the total set of 14,243 genes which passed quality control) (Figure 3C).

Gene set enrichment analysis (GSEA, [34]) performed against the hallmark gene set collection [53] of the Molecular Signatures Database (MSigDB) revealed that siZNF cells exhibit transcriptomic hallmarks of MYC oncogene activation (exact data on MYC gene expression to be found in Appendix A), metabolic shift towards oxidative phosphorylation and activation of stress response (DNA repair, unfolded protein response) (Figure 3D). Interestingly, although the analyzed cell line (HCT116 p53 KO) is deficient in p53, the analysis revealed up-regulation of genes involved in p53 pathways and networks. This is a broad group of genes related e.g., to stress response, DNA repair, as well as regulation of growth/apoptosis, angiogenesis and metastasis.

Notably, in the same analysis epithelial to mesenchymal transition received negative normalized enrichment score (NES), indicating down-regulation of related genes. Moreover, significant enrichment (with negative NES) of such categories as E2F targets (cell cycle related targets of E2F transcription factors), G2/M checkpoint and mitotic spindle, indicate aberration of cell-cycle gene expression in siZNF cells. The network of cell cycle related genes is largely inhibited in cells deficient in siZNF84, treated with Dox (Appendix A).

GSEA analysis against the more detailed REACTOME database recapitulated the main findings obtained with MSigDB hallmark gene set, pointing to the increased expression of genes linked to response to stress (such as heat stress and DNA damage) in siZNF cells, decreased expression of genes participating in the regulation of the cell cycle, as well as up-regulation of the genes influencing the energy state of cells (related to oxidative phosphorylation, mitophagy and mitochondria biogenesis) (Appendix A, Figure 3E and Appendix A). Intensive protein production may occur in siZNF cells as they exhibit elevated level of translation-related transcripts and this might explain the increased demand for energy, redirecting them towards oxidative phosphorylation (Figure 3D). Interestingly, cells deficient in ZNF84 seem to have reduced cell death signaling in comparison with control (Appendix A).

Overall, transcriptomic data suggest that siZNF cells cultured in the presence of genotoxic drug mobilize their stress-defense mechanisms (including rewiring of metabolic pathways), and in the next step this may be associated with altered stress resistance and modified oncogenic potential (changes in expression in MYC and p53 networks, inhibition of genes related epithelial to mesenchymal transition), yet the direction of changes requires further studies. The implications of low ZNF84 level for cancer cells (e.g., their aggressiveness and treatment response) is an important question.

Interestingly, we observed that altered expression of genes related to e.g., cell cycle, translation and oxidative phosphorylation also occurs in cells that were siRNA-transfected but not treated with Dox (Marked as “Untreated” in Appendix A; RNA-seq data from these cells presented also in Appendix A–F)). This indicates that the role of ZNF84 is not limited to therapy-evoked stress conditions but can affect vital cellular processes also in untreated cells. Therefore the role of this protein can reach beyond chemotherapy-related context, which may be an interesting direction for future research.

Since the previously described high-throughput transcriptomic analysis (Figure 3D and Appendix A) indicated deregulation of cell-cycle related genes, we analyzed the expression of genes that have been shown to be periodically expressed in the human cell cycle [54]. We observed that M/G1 genes (active in late M and early G1 phase) showed a tendency for up-regulation in siZNF cells, in contrast to G1/S and G2/M phase genes (Figure 4A). This suggests that siZNF cells are characterized by an increased population of mitotic and early G1 cells. This can be caused, among other reasons, by mitotic defects, such as prolonged duration of mitosis. Transcriptomic analysis (Figure 3D) indicated inhibition of expression of mitotic spindle genes. Concordantly, analysis of expression patterns of genes previously described to be down-regulated in aberrant mitosis [55] (Figure 4B) indicated their prevailing down-regulation also in siZNF cells (however no statistical significance was reached in the GSEA analysis for this small gene set). We performed a cytometric analysis of anti-MPM-2-stained cells in order to evaluate the mitotic fraction in siNeg and siZNF cell populations. As the presence of cytostatic drug (Dox) inhibits progression of the majority of cells into mitosis, we applied nocodazole (200 ng/mL) treatment simultaneously with Dox for 4 h in order to accumulate mitotic cells for the analysis. We observed an increased fraction of mitotic cells in siZNF cells compared to siNeg (Figure 4C). This observation can enforce the hypothesis about aberrant (prolonged) mitosis, which emerged from transcriptomic studies.

Another possible explanation of increased mitotic fraction is obviously the enhanced proliferation (overriding growth arrest). This would be consistent with the results of BrdU assay, which indicated increased DNA synthesis in siZNF vs. siNeg cells (analyzed on 3rd day of culture with doxorubicin). In order to gain more insight in cell cycle deregulation, we performed cytometric analysis of DNA content. We did not detect significant increase in S-phase (proliferating) cells by PI cell cycle analysis, but probably more accurate method, based on pulse BrdU combined with PI staining should be used to this aim, to provide reliable data.

The most striking effect of ZNF84 down-regulation on the cell cycle was an increased fraction of cells with 2N DNA content (Figure 4D,E), attributed to G1 phase and not to G0, as cells stained positive for Ki67 (Appendix A). We also observed an increase in 2N cells in H358, H1299 and HCT116 p53WT cell lines (Figure 4E and Appendix A).

In most cell lines the shift to G1 was attained mainly at the expense of tetraploid (G2/M) cells (Appendix A), indicating an abrogation of p21-driven, p53-independent G2/M arrest by ZNF84 deficiency. By contrast, the biggest change observed in HCT116 p53KO cells was the reduction of sub-G1 (<2 N) and polyploid (>4 N) fractions, indicating reduced cell death and inhibited polyploidization. This, again, suggests activation of cytoprotective and genomic stability-promoting mechanisms, partially corroborating the results of transcriptomic analyses. As polyploidization often accompanies senescence [56], the reduction in the number of polyploid cells is consistent with attenuation of senescence in siZNF cells. 

Taken together, we observed that in the prompt response to doxorubicin (24 h of treatment) cells deficient in ZNF84 more often than their siNeg counterparts remain in the diploid state, which can be caused by prolonged mitosis, decreased susceptibility to cell death, inhibited polyploidization and abrogation of G2/M arrest.

### 3.3. ZNF84 Level Impacts Senescence vs. Proliferation Outcome of Genotoxic Treatment and Affects Patient Survival

Given the fact that senescence induction takes place also *in vivo* in cancer patients undergoing chemo- or radiotherapy, and that it contributes to the disease outcome, investigation of senescence-related mechanisms is of clinical importance. As we have observed that upon ZNF84 silencing many cancer cell lines exhibit cell cycle perturbation, depletion of some markers of senescence and, perhaps most intriguingly, decreased DNA damage signaling (Figure 2B), we were interested how ZNF84 deficiency would affect cancer cell fitness. 

To mimic the conditions of chemotherapy, in which the drug is cleared from the organism by metabolic processes and secretory system, we refrained from culturing cells in constant presence of Dox for 5 days. Instead, we introduced pulse treatment, in which Dox-containing medium was removed after 24 h and cells received new, drug-free medium for the remaining time of the experiment. We have used such an experimental scheme before [57] and we observed that senescence markers developed extensively in this model, however after 4 to 5 days in drug-free medium proliferating cells re-emerged, indicating escape from senescence. Here, we observed that the fraction of proliferating cells is potentiated approximately 3-fold by silencing ZNF84, as indicated by colony formation assay (Figure 5A). This suggests that ZNF84 deficiency potentiates senescence escape/senescence bypass (which are twin phenomena that may only be distinguished by single-cell studies). Interestingly, increased proliferative potential appeared also in p53WT cells (Figure 5A), with almost the same magnitude as in p53-deficient cells, pointing to a more universal mechanism, not restricted to substituting p53’s role by ZNF84.

To examine if ZNF84 expression impacts on disease outcome, we analyzed TCGA data from 275 colon cancer patients and found a strong correlation between ZNF84 expression and overall survival (Figure 5B). Interestingly, low ZNF84 expression was associated with increased survival. This is contrary to our expectation that it would be low level of ZNF84 that is to be associated with chemoresistance and poor prognosis. However, caution must be taken when trying to compare *in vitro* and clinical data, especially in the case of a protein, which has not yet been a subjected to many studies, not to mention the specific conditions of the *in vitro* study, such as p53-deficient background, context of specific (doxorubicin) cytostatic treatment and non-physiological protein level (due to gene knock-down).

We believe this is an interesting field for further research, focused e.g., on identification of possible presence of negative feedback loops between p21 and ZNF84, affecting the levels of these proteins in long-term conditions. In summary, from the results presented here, a model emerges of ZNF84 as a protein impacting cancer cells’ response to chemotherapy and affecting cancer progression. 

## 4. Discussion

The transcription factor p53, its downstream target p21, and the protein p16INK4 (the latter two being potent cell cycle inhibitors) constitute the major axis in the activation of senescence program [58]. Yet, their activity is not a strict prerequisite for senescence, as in cells lacking any one of them, senescence could still be induced [1]. Senescence of p53-deficient cells has been long known as a phenomenon, and elevated expression of p21 in those cells was described repeatedly, together with various transcriptional and post-transcriptional regulators [26,27]. Yet, the subject is far from being exhausted and we still lack the comprehensive picture, of how p21 up-regulation is achieved in various cellular contexts and upon different stimuli. 

In this study, we were interested to investigate how senescence is regulated in the absence of p53 and p16. Our proteomic approach allowed us to look for novel players involved in p21 control in an unbiased manner, leading us to the identification of ZNF84. Consecutive siRNA-based studies confirmed attenuation of *CDKN1A* gene transcription and p21 protein production in ZNF84 deficient cells. Although ZNF84 identification method, based on chromatin pulldown followed by mass spectrometry, was designed to detect direct influence on target gene (*CDKN1A*) by physical presence on its promoter region, we cannot exclude also indirect impact of ZNF84 on p21. For example, upregulation of MYC was detected in our transcriptomic data from ZNF84-deficient cells, and MYC is a well-established *CDKN1A* repressor [59]. Interestingly, ZNF84 is a poorly characterized protein that has never been associated with cell cycle, DNA repair or senescence, prompting us to investigate its function in more detail. p21 is regarded as a key senescence mediator [1,26], since without it human fibroblasts bypass senescence and fail to arrest the cell cycle in response to DNA damage [19]. Depletion of ZNF84 not only inhibited expression of *CDKN1A*, but also conferred phenotypic characteristics of perturbed senescence, such as increase of DNA synthesis, depletion of SA-β-Gal activity, reduction of DNA damage signaling, decrease of the size of cell nuclei, and perturbations of cell cycle arrest. Thereby, this places ZNF84 among the players at the top of the hierarchy to regulate induction of senescence in p53-deficient background.

An important feature of senescent cells is the presence of DSBs, which results in the activation of the DDR cascade, maintaining senescence [60]. Therefore, decreased γH2AX signaling observed in cells with down-regulated ZNF84 is in accord with the reduction in other markers of senescence in these cells. Nevertheless, the mechanism by which ZNF84 is linked to DNA integrity requires further elucidation. We wondered whether reduced expression of p21 in the cells with silenced ZNF84 could directly translate to the decreased H2AX phosphorylation. Mechanistically, in the cellular response to DSBs, ATM kinase is activated to phosphorylate H2AX [61] as well as p53, which then trans-activates *CDKN1A*. As such, H2AX phosphorylation is independent of both p53 and p21. Beyond that, the relation between DNA repair and p21 has been analyzed in several previous studies, but results are contradictory. While some experiments showed that p21 is required for DNA repair in cancer cells [62,63,64] and fibroblasts [65], others reported that p21 is not necessary [66,67] or even that it inhibits nucleotide excision repair [68] and interferes with trans-lesion DNA synthesis [69]. The differences may be caused by the used model cell line, type of repair mechanisms, as well as type and extent of DNA damage. Therefore, there is lack of straightforward link between ZNF84, p21 and the number of DNA damage foci. Nevertheless, differential expression data, obtained by RNA-sequencing, pointed to up-regulation of genes involved in the DNA repair, especially in double strand break response in siZNF cells (Figure 3D,E). Noteworthily, among the up-regulated genes belonging to the DNA damage repair category there is EYA3 gene, encoding a tyrosine phosphatase of very intriguing function. It promotes efficient DNA repair rather than apoptosis in response to genotoxic stress, by executing a damage-signal-dependent dephosphorylation of carboxy terminal tyrosine phosphate (Y142) from γH2AX [70]. In other words, when Y142 phosphorylation is removed by EYA3, binding of repair factors to γH2AX is enhanced. This could be one explanation why ZNF84-depleted cells are able to survive despite constant presence of genotoxic drug, and why they exhibit less damage foci than those cells where ZNF84 was not silenced. We consider this a promising direction for future research, focused on clarifying the apparent genoprotective effect evoked by ZNF84 silencing. Obviously, the reduced number of γH2AX foci can also be a result of multiple other mechanisms, not only by increased effectiveness of repair, e.g., by an increased removal of a genotoxic drug from cells or by reduced detection of the damage by DDR proteins.

DNA synthesis, measured by BrdU assay, was another marker of senescence that was affected in ZNF84-depleted cells. The increased fraction of BrdU-positive cells could be a result of decreased level of p21 cell cycle inhibitor and consequent redirecting cells towards proliferation, even in the presence of genotoxic stress. Such effect was shown in HCT116 p21KO cells treated with doxorubicin [1]. More relevant to our model, proliferation can also be evoked by the presence of low concentrations of p21 protein (not a complete knock-out), through the assembly and activation of cyclin D/Cdk4 or Cdk6 complexes, as shown in cancer cell lines [71,72]. This means that p21 does not always inhibit proliferation, depending e.g., on concentration. Concordantly, Weiss [73] considered the term “cell cycle inhibitor” in the description of p21 a “confusing misnomer”. Galanos et al. also reported that p21 could have oncogenic properties and lead to bypassing senescence [74]. Recently, single-cell analysis showed that proliferation-fated cells (contrary to senescence-fated ones) exhibit elevation of p21 to the intermediate level during the 24-h treatment with doxorubicin [75]. We used bulk cell populations, but in the light of this research it is possible that siZNF cells which exhibited increased incorporation of BrdU were proliferation-fated cells, in which p21 reached intermediate level. This could occur, for example, due to incomplete silencing of ZNF84 resulting from the nature of gene silencing with siRNA.

Nevertheless, in the interpretation of increased BrdU incorporation as an indicator of increased proliferation, caution must be taken. Unexpectedly, when we performed cell cycle analysis (DNA content), we did not observe substantial increase in the percentage of S phase cells. So even though time-points of BrdU and cell cycle analysis were different, it is possible that DNA replication may not be the main cause of the increased BrdU incorporation. Interestingly, BrdU incorporation can also be associated with DNA repair process, especially in p53-deficient cancer cells [76]. This may (at least partially) explain the increased BrdU incorporation in siZNF cells and supports the hypothesis about enhanced DNA repair in ZNF84-deficient cells.

In the DNA content analysis we observed an accumulation of ZNF84-depleted cells in G1 phase—which is intriguing, taking into account the fact that G1 arrest is known as p21- and p53-dependent [77,78]. However, G1 accumulation can also be considered an effect of increased G1 phase length, which occurs in a p21-independent manner [77]⁠. Importantly, Hsu et al. [75] showed also that cells exhibiting proliferation fates upon doxorubicin treatment (like siZNF cells in our colony formation assy) typically had a long G1 phase during the drug pulse and resumed cell cycle after drug removal. As it was established recently [79], one of the mechanisms of G1 phase extension is by inactivation of CDK4/6, which can occur due to p21 deficiency (as mentioned previously, CIP/KIP CDK inhibitors bind to and activate cyclin D-CDK4/6 complexes, promoting proliferation [71]). Concordantly, CDK4/6 synthetic inhibitor PD0332991 (Palbociclib) slowed down G1 progression of HCT116 cells treated with doxorubicin, potentiated senescence-escape and enhanced colony formation potential [80]. Whether siZNF HCT116 p53KO cells exhibit p21-deficiency-related G1 phase delay or other mechanisms cause the increase of G1 cells, requires further study. 

Abrogation of cell cycle control in siZNF cells was indicated not only by an increase in the fraction of G1 cells, but also by increased fraction of mitotic (MPM-2 positive) cells. Of note, deficiency of p21 has been reported to prolong the duration of mitosis by extending metaphase, anaphase and cytokinesis [81]. 

RNA-seq analysis indicated various alterations in cells deficient in ZNF84. Apart from deregulation of cell cycle, transcriptome analysis showed e.g., inhibition of epithelial to mesenchymal transition. This could be an effect of p21 deficiency, which was associated previously with the compromised migration and invasion capability of various trophoblastic and cancer cell lines (reviewed in [82]). Our transcriptomic studies indicated also that siZNF cells exhibit increased transcription of genes involved in oxidative phosphorylation, whereas during normal senescence the overall ATP production by oxidative phosphorylation is reduced (reviewed in [83]). This corroborates the results of biochemical analyses (Figure 2) showing that ZNF84 depletion counteracts senescence. The alteration of energy metabolic profile towards oxidative phosphorylation comprises a reversal of the so-called Warbug effect (a particular metabolic phenotype exhibited by many rapidly growing tumours, characterized by reduced oxidative phosphorylation and enhanced glycolysis; adaptation to intra-tumour hypoxia) [84]. Hypothetically, increased oxidative phosphorylation could be a transient state, used for a rapid response to stress conditions (doxorubicin) providing cells with energy for intensive protein production (also indicated by the RNA-seq analysis), enabling survival and resumption of proliferation.

Finally, we were interested whether the alterations evoked by ZNF84 down-regulation could be relevant from a clinical perspective. We found that high expression of ZNF84 was associated with shorter survival of colon cancer patients. Accordingly, a previous study reported that ZNF84 expression in cervical cancer samples was higher than in surrounding healthy tissue—and that it correlated with tumor size [85]. These results appear discrepant to our *in vitro* studies. Obviously, clinical data come from a seemingly different setting than the results of experiments, where we analyzed the response to chemotherapy and ZNF84 was removed before the treatment and prevented p21 up-regulation. In this case low level of ZNF84 decreased the cytostatic (senescence) effect of the treatment. In order to pursue the role of ZNF84 in cancer, single cell analysis of ZNF84/p21 level together with cell fate tracing would be desirable.

The future perspectives of the discovery may be relevant to the clinical applications. Although low level of ZNF84 seems to impede the cytostatic effect of anticancer treatment (namely senescence), paradoxically it can be attempted to be exploited in therapy, with modified/innovative approach. According to recent studies, residual senescence, occurring as an off-target effect of anti-cancer therapy in less accessible sites of the tumour (where drug concentration is low), is often source of aggressive tumour relapse. By inhibiting ZNF84 concomitantly with applying chemotherapy to cancers with non-functional p53 which rely on ZNF84 for p21 up-regulation, we would counteract senescence induction. Potentially such cells could be more easily targeted and killed with a next round of chemotherapy than dormant, apoptosis-resistant senescent cells. In such scenario, patients with p53-deficient tumours could receive combined therapy of ZNF84 inhibitor with chemotherapy to increase long-term effectiveness of the treatment.

## Figures and Tables

**Figure 1 cancers-13-02115-f001:**
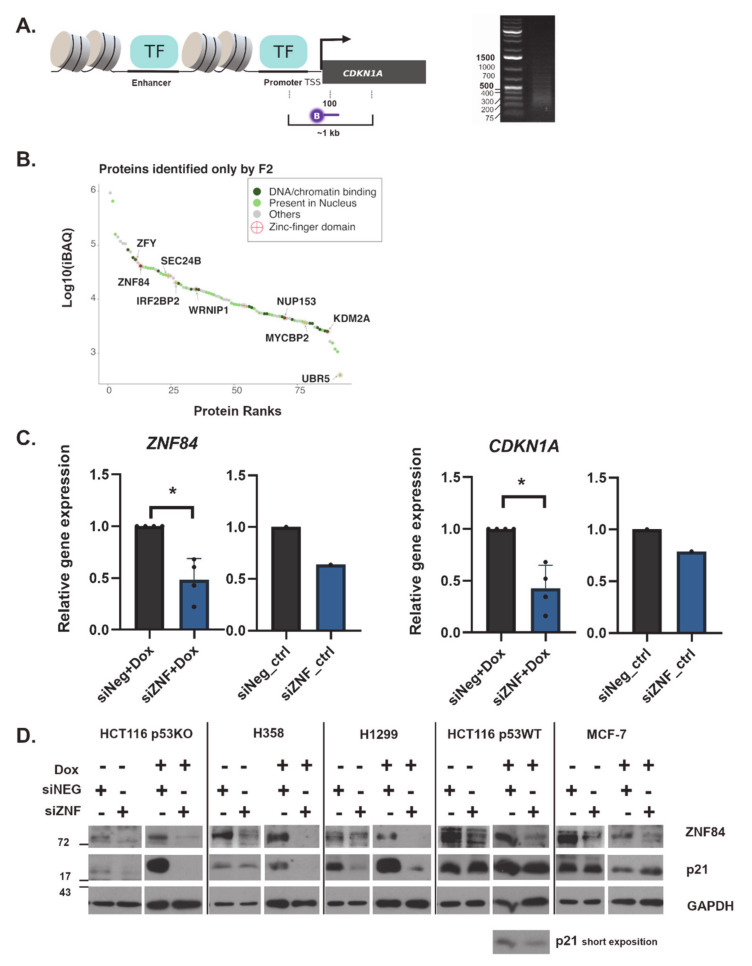
ZNF84 identified by proteomic approach plays role in the regulation of p21. (**A**) Left: Scheme 100. b.p. downstream of *CDKN1A* transcription start site (TSS). Right: Image of agarose gel electrophoresis visualizing the range of DNA fragments, generated by sonication (5 cycles of 30 sec ON, 30 sec OFF); image cropped to show only DNA ladder and the relevant lane; (**B**) Proteins specifically identified using *CDKN1A*-targeting probe ranked according to their iBAQ intensities. DNA or chromatin-binding proteins are dark green. Proteins present in nucleus are pale green. Proteins that do not fall in the aforementioned categories are gray. Proteins that contain a zinc-finger domain are labelled with a red cross in circle; (**C**) RT-PCR analysis of *ZNF84* (graphs on the left) and *CDKN1A* (graphs on the right) mRNA in HCT116 p53KO cells upon transfection with control siRNA (siNeg, gray bars) or ZNF84-targeting siRNA (siZNF, blue bars) and either a 2-day treatment with doxorubicin (Dox) or lack thereof (ctrl). The mean of raw Ct values for *GAPDH* was subtracted from the mean Ct for *CDKN1A* or *ZNF84,* giving ∆Ct valuesand then ∆∆Ct was calculated as a difference between ∆Ct for siZNF and ∆Ct for siNeg-transfected cells. The fold gene expression value was calculated as 2 to the power of negative ∆∆Ct and the value was plotted in the graph. Bars represent mean values ± SEM, black dots represent values from individual experiments, * indicates *p*-value < 0.05 (statistical significance calculated on raw Ct data, with paired one-tailed t-test upon distribution normality check with Shapiro-Wilk method); (**D**) The levels of ZNF84 and p21 proteins in control and Dox-treated cells, analyzed by western blotting. Timepoints for each cell line were selected according to p21 kinetics: p53-deficient cells (HCT116 p53KO, H358 and H1299) were analyzed after 2 days of Dox treatment whereas p53-proficient cell lines (HCT116 p53WT, MCF-7) after 1 day of Dox. GAPDH was used as a loading control. Representative images of at least 3 independent experiments per each cell line. For HCT116 p53WT cells the additional image from short exposition of p21 is shown below.

**Figure 2 cancers-13-02115-f002:**
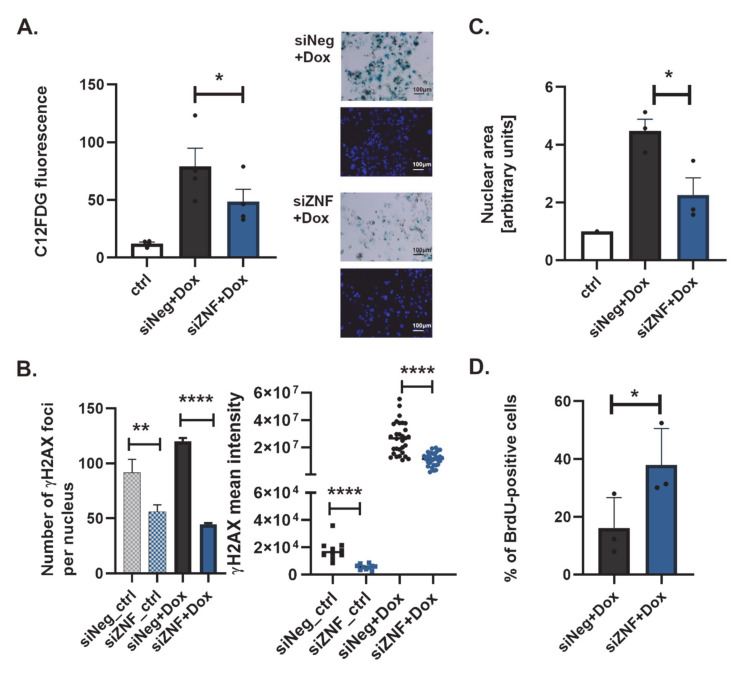
ZNF84 downregulation causes attenuation of senescence in HCT116 p53KO cells. White bars represent data from untreated and non-transfected HCT116 p53KO cells, gray non-patterned bars—cells transfected with control siRNA (siNeg) and treated with 100 nM doxorubicin, blue non-patterned bars-cells transfected with ZNF84 siRNA (siZNF) and treated with 100 nM doxorubicin: (**A**) SA-β-Gal activity (cytometric analysis). Bars represent mean values ± SEM, black dots—values from individual experiments. Paired two-tailed t-test upon distribution normality check with Shapiro-Wilk method: * *p*-value < 0.05; On the right side representative images from cytochemical analysis of SA-β-Gal activity. Magnification 100x; (**B**) Immunofluorescent detection of γH2AX DNA damage foci. Unpaired two-tailed t-test: ** *p*-value < 0.01, **** *p*-value < 0.0001 (Graph on the left: Numbers of foci analyzed throughout the *z*-axis in at least 50 DAPI-stained nuclei from each of the three independent experiments with Dox, mean values plotted with SEM error bars. Graph on the right: intensity of γH2AX fluorescence was analyzed in the total number of 30 nuclei from three independent experiments with Dox, dots represent individual values, horizontal lines indicate mean of these values; data from untreated cells presented as patterned bars and squares) (**C**) Analysis of the size of DAPI-stained cell nuclei. The areas of at least 50 cells from each of the three independent experiments were analyzed, mean values were plotted, in arbitrary units relative to control, with SEM error bars. Paired one-tailed t-test upon distribution normality check with Shapiro-Wilk method: * *p*-value < 0.05; (**D**) Analysis of BrdU immunofluorescent detection. The percentage of BrdU-positive cells was calculated in the total cell population of at least 200 cells per each of the three independent experiments and plotted as black dots; bars represent mean values ± SEM; Paired one-tailed t-test upon distribution normality check with Shapiro-Wilk method: * *p*-value < 0.05.

**Figure 3 cancers-13-02115-f003:**
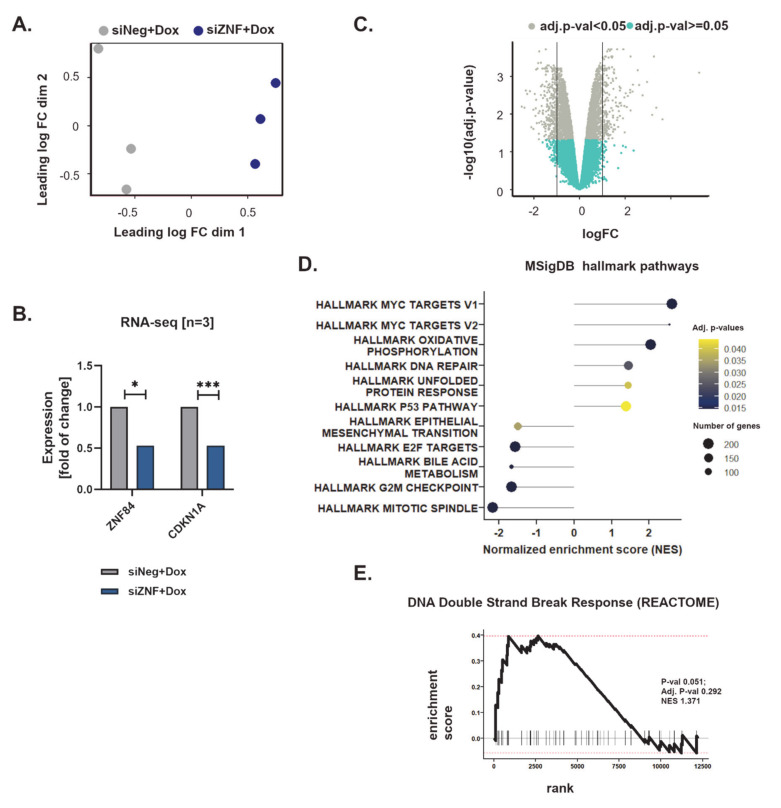
Analysis of transcriptomic changes in doxorubicin-treated HCT116 p53KO caused by ZNF84 downregulation. RNA-seq analysis was performed on cells treated with doxorubicin for 2 days, *n* = 3. (**A**) Visualization of a principle components analysis (PCA), which determines the greatest sources of variation in the data. Each point represents an RNA-Seq sample. Values grouped by the type of siRNA used for the transfection (gray: siNeg + Dox, blue: siZNF + Dox); (**B**) Expression data (fold of change) of selected genes: *ZNF84* and *CDKN1A* in cells transfected with either siNeg (gray) or siZNF (blue). * indicates adj. *p*-value < 0.05, *** adj. *p*-value < 0.001; (**C**) Volcano plot representing differential expression of genes between siNeg + Dox and siZNF + Dox samples (*x*-axis: logarithm of fold change, *y*-axis: negative decimal logarithm of FDR-controlled *p*-value); significantly differentially expressed genes (adj. *p*-value < 0.05) represented by gray points; (**D**) Results of Gene Set Enrichment Analysis (GSEA) of the differentially expressed genes assigned to Hallmark gene sets in MSig database. Bars represent values of Normalized Expression Score (NES) with significant enrichment (adj. *p*-value < 0.05); (**E**). Representation of GSEA analysis of the differential expression data of genes which play role in the DNA Double Strand Break Response, as defined in the REACTOME database.

**Figure 4 cancers-13-02115-f004:**
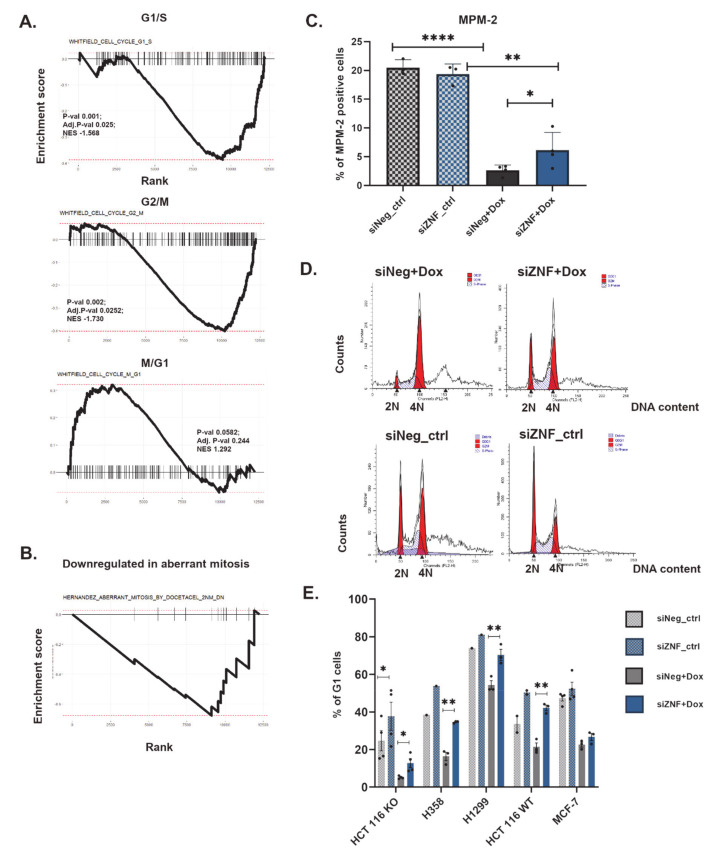
Cell cycle is deregulated in ZNF84 knock-down cells treated with doxorubicin. (**A**) Representation of GSEA analysis of the differential expression data of genes defined by Whitfield et al. (Whitfield et al. 2002) as characteristic to G1/S (upper graph), G2/M (central graph) and M/G1 (lower graph); (**B**) Representation of GSEA analysis of the differential expression data of genes identified by Hernández-Vargas et al. (Hernández-Vargas et al. 2007) to be downregulated in aberrant mitosis; (**C**) Percentage of mitotic (MPM2-positive) HCT116 p53KO cells. Bars represent mean values ± SEM in untreated (ctrl, patterned bars) as well as treated with 100 nM doxorubicin for 24 h (+Dox, plain bars) upon transfection with either control siRNA (siNeg, gray bars) or ZNF84 siRNA (siZNF, blue bars). Black dots represent values from individual experiments. Paired one-tailed t-test upon distribution normality check with Shapiro-Wilk method: * *p*-value < 0.05, ** *p*-value < 0.01, **** *p*-value < 0.001; (**D**) Representative histograms of flow-cytometric analysis of DNA content in HCT116 p53KO cells treated with 100 nM doxorubicin for 24 h upon transfection with either control siRNA (left-hand side) or ZNF84 siRNA (right-hand side); histograms from cells non-treated with the drug presented below; (**E**) Percentage of G1 cells, analyzed by flow cytometry of PI-stained HCT116 p53KO, H358, H1299 and HCT116 p53WT. Bars represent mean values ± SEM in untreated (ctrl, patterned bars) as well as treated with 100 nM doxorubicin for 24 h (+Dox, plain bars) upon transfection with either control siRNA (siNeg, gray bars) or ZNF84 siRNA (siZNF, blue bars). Black points represent values from individual experiments. Paired one-tailed t-test upon distribution normality check with Shapiro-Wilk method: * *p*-value < 0.05, ** *p*-value < 0.01.

**Figure 5 cancers-13-02115-f005:**
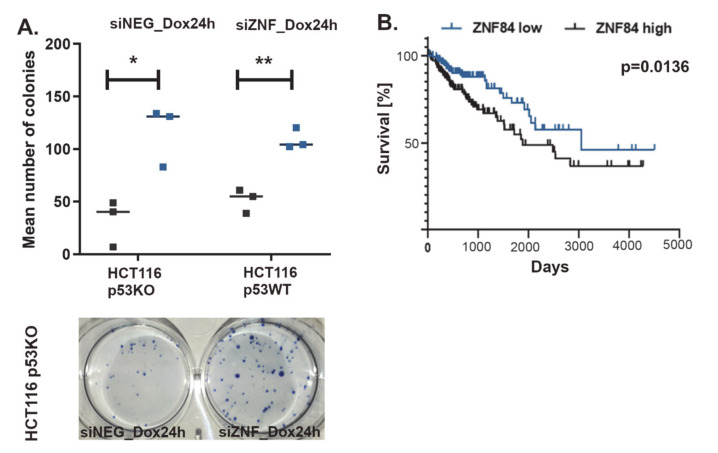
ZNF84 level affects both proliferative potential of HCT116 p53KO cells and colon cancer patient survival. (**A**) Results of colony formation assay. Cell were transfected either with control siRNA (black points on the graph) or with ZNF84 siRNA (blue points) before administering pulse of doxorubicin (24 h) and further drug-free culturing. Points represent numbers of colonies formed by HCT116 p53KO as well as HCT116 p53WT cells (mean values from three technical replicates). Horizontal lines represent mean values of three independent biological replicates. Paired one-tailed t-test upon distribution normality check with Shapiro-Wilk method: * *p*-value < 0.05, ** *p*-value < 0.01. Picture of a culture dish below the graph; (**B**) Kaplan-Meyer plot depicting survival rates of colon cancer patients exhibiting either high (black line, *n* = 143) or low (blue line, *n* = 140) expression of ZNF84.

## Data Availability

The mass spectrometry proteomics data have been deposited to the ProteomeXchange Consortium via the PRIDE [86] partner repository with the dataset identifier PXD022542. The transcriptomics data have been deposited to the Array Express with the dataset identifier E-MTAB-9848.

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
