# Peer review of "Chromatin-Directed Proteomics Identifies ZNF84 as a p53-Independent Regulator of p21 in Genotoxic Stress Response"

_cancers, 2021, doi:10.3390/cancers13092115_

Round 1

Reviewer 1 Report

The revised manuscript has been significantly improved and should be accepted for publication.

I thank the authors for their time and effort in carefully responding to my requests.

I would recommend to include some of the revision figures into the manuscript (Rev1. Fig.1, Rev1. Fig.4, Rev1. Fig.5 and Rev1. Fig. 6) or at least as Suppl. Information, as they provide important additional information.

Author Response

We would like to thank the Reviewer for the effort to review the revised version of the manuscript and for the contribution in refining the publication.

According to the suggestion of the Reviewer, the following revision figures were added to the manuscript:

  • Fig.1 was added as a point B into the new Suppl. Fig. 1 (resulting in a change in the numbering of the remaining supplementary figures);
  • Fig.4 was added as a new point A and D to the new Suppl. Fig.3 (former Suppl. Fig. 2; this modification changed the order of the remaining elements in this figure; additionally, plots from the former Suppl. Fig. 4 were transferred here);
  • Fig.5 was added as a point C into the new Suppl. Fig. 1;
  • Fig. 6 was added as a new point A into the new Suppl. Fig. 1.

Reviewer 2 Report

The article by Strzeszewska-PotyraÅ‚a et al. entitled “Chromatin-directed proteomic identifies ZNF84 as a p53-independent regulator of p21 in genotoxic stress response” describes the potential role of this newly identified zinc finger domain protein in the induction of the CDK inhibitor p21 after DNA damage. The CDK inhibitor p21 is known to be induced by DNA damage to halt cell cycle progression and known to be induced when cells undergo senescence after DNA damage in order to maintain cell cycle arrest. The authors show that depletion of ZNF84 results in decreased induction of p21 mRNA after doxorubicin treatment. They also show reduced p21 protein levels after DNA damage in cell treated with siZFN compared to siCTRL in various cell lines. This was the case in multiple p53-KO cell lines, but not the case in p53-WT cell lines. The authors also showed reduced senescence markers upon depletion of ZNF84. Interestingly, the authors also showed reduced H2AX foci in cells lacking ZNF84. The authors also show less convincing evidence regarding some cell cycle effects upon ZNF84 depletion and some RNAseq data showing genes involved in various pathways were affected. Finally, the authors show some analysis of cancer survival in patients with high/low ZNF84. Overall the manuscript is clearly written, the methods are well described, however the article is longer than it needs to be and some of the conclusions are not supported by the data presented. This is the second version of the manuscript and the authors have added some data, though not all data I requested initially and the improvement to the work is marginal at best. I would request the authors to address the following questions prior to publication:

Major points

  1. The authors have added the siCTRL and siZFN (without Dox) samples I requested, however, the methods or legends don’t specify how are those samples set to 1 by default. It’s unclear what are they relative to? According to the data presented and they way it’s presented treatment of siCTRL cells with Dox does not result in p21 expression? Or, if it does we can’t see it due to the way the data is presented. The data should be presented siCTRL (-Dox) vs siCTRL (+Dox) and the same regarding siZFN so that we can see that there’s an induction in p21 that does not occur in the siZFN cell line.
  2. The data is a bit more clear regarding the western blots but if the authors want to make the case this is a transcriptional defect, they must show transcription data.
  3. The same issue arises on Fig. 2: the authors show a “ctrl” (which is really unknown as to what does it represent) and no siNeg (-Dox) or siZFN (-Dox). The siCTRL (-Dox) vs siCTRL (+Dox) must be presented for Fig. 2A, C, D.
  4. 2B also presents some problems. First, the basal level of H2AX for untreated cells is way too high (~90 foci/cell?) and Dox treatment only marginally increases the number of foci, which doesn’t make sense. The data regarding foci intensity makes more sense.
  5. 2D: Once again the authors don’t show the controls (siCTRL (-Dox) vs siCTRL (+Dox)­) and that data is required. Moreover, the data showing that there’s increased BrdU incorporation after DNA damage suggest more a checkpoint defect than a senescence phenotype. Interestingly, and later noted by the authors, checkpoints are among the downregulated pathways.
  6. The authors refer multiple times to the amount of cells in S phase by using flow cytometry and PI, which is not appropriate. BrdU must be used, and also the cell cycle plots only show the treated samples, no controls (siCTRL (-Dox) vs siCTRL (+Dox) and similar for siZFN) are shown.
  7. Controls are also missing in Fig. 5
  8. There’s very little information in this manuscript for the characterization of this new protein that the authors say, very little is known about. The authors should make an effort to characterize the cells lacking this protein and that means that the siCTRL (-Dox) vs siZFN (-Dox) must be included in all experiments.

Minor points:

  1. Name of genes on Fig. 1B are not legible, need to be enlarged
  2. 3: D the figure is not legible

Author Response

We would like to thank the Reviewer for the effort to carefully review the revised version of the manuscript and for all the inspiring comments and questions.

We are glad to provide the answers – please see the attachment containing the point-by-point response.

Round 2

Reviewer 2 Report

Authors have addressed my concerns.

This manuscript is a resubmission of an earlier submission. The following is a list of the peer review reports and author responses from that submission.

Round 1

Reviewer 1 Report

The paper by Strzeszewska-PotyraÅ‚a et al. entitled “Chromatin-directed Proteomics Identifies ZNF84 as a p53-2 independent Regulator of p21 in Genotoxic Stress Response” does fit within the scope of the journal. They identified ZNF84 as a trans-activator of CDKN1A facilitating senescence bypass in response to doxorubicin.

The manuscript is well written. Most of the results are sound and convincing. However, a crucial experiment is missing to provide the direct connection/proof whether ZNF84 is a direct activator of CDKN1A transcription. Otherwise, it could be an indirect mechanism, as ZNF84 downregulation leads to Myc overexpression, which is a known p21 repressor.

Additional experiments are needed to prove that ZNF84 is a direct activator of CDKN1A transcription:

  • Luciferase Reporter Assay with the p21-Promoter
  • Figure 4: Polyploidy and mitotic errors should be counted via immunofluorescence staining.

Missing controls:

  • Figure 1 C and D: all experiments are done with dox. I am missing the controls without dox. siNeg and siZNF treated with the solvent control (DMSO?)
  • Figure 1C: Please check for myc expression.
  • Figure 1D: Please provide the Western blot staining of p53.
  • Figure 4: all experiments are done with dox. I am missing the controls without dox. siNeg and siZNF treated with the solvent control (DMSO?)

Open questions:

  • For siRNA studies, generally at least 2 different siRNAs or a pool of several siRNAs are used to exclude off-target effects. Did the authors use only one sequence? Please comment.
  • And if yes, I would recommend to verify some of the data with a second siRNA sequence.
  • Moreover, a knockdown of 50% with siRNA at 48 h is not the best result. Have you checked the efficacy of your siRNA in the long-term experiments? Usually, the siRNA knockdown is not stable over such a long time.

Additional comment:

Line 705: The authors could mention the work by Galanos P et al. (Chronic p53-independent p21 expression causes genomic instability by deregulating replication licensing. Nat. Cell Biol. 2016;18:777–789. doi: 10.1038/ncb3378) showing that sustained p21 expression also leads to bypassing senescence and chemoresistance.

Reviewer 2 Report

The article by Strzeszewska-PotyraÅ‚a et al. entitled “Chromatin-directed proteomic identifies ZNF84 as a p53-independent regulator of p21 in genotoxic stress response” describes the potential role of this newly identified zinc finger domain protein in the induction of the CDK inhibitor p21 after DNA damage. The CDK inhibitor p21 is known to be induced by DNA damage to halt cell cycle progression and known to be induced when cells undergo senescence after DNA damage in order to maintain cell cycle arrest. The authors show that depletion of ZNF84 results in decreased induction of p21 mRNA and also protein levels after DNA damage. This was the case in multiple p53-KO cell lines, but not the case in p53-WT cell lines. The authors also showed reduced senescence markers upon depletion of ZNF84. Interestingly, the authors also showed reduced H2AX foci in cells lacking ZNF84. The authors also show less convincing evidence regarding some cell cycle effects upon ZNF84 depletion and some RNAseq data showing genes involved in various pathways were affected. Finally, the authors show some analysis of cancer survival in patients with high/low ZNF84 and high/low p21. Overall the manuscript is well written, the methods are well described and the conclusions are supported by the data presented. I would request the authors to address the following questions prior to publication:

  • Zinc finger domain proteins are known to function in multiple processes such as transcription and DNA repair. I would like the authors to speculate where and how they believe this protein works and how it could potentially affect the transcription of p21.
  • Given that ZNF84 is rather uncharacterized the authors should include the data of the cellular effects that the downregulation of ZNF84 alone has (without doxorubicin treatment) on the different assays. This would be particularly interesting in the case of H2AX foci. The authors show that H2AX foci are reduced in siZNF84+Dox compared to siNeg+Dox, but the comparison without Dox is not shown. If there’s higher levels of H2AX foci in siZNF84 it could suggest that this protein could play a role in DNA repair. If H2AX is lower under both conditions (+/- Dox) it could suggest that this protein may play a role in DDR signaling.
  • 11 Ln. 469 Also, related to H2AX and Fig. 2B. The authors write that the lower H2AX foci levels “pointing to reduced DNA damage levels in these cells.” This is not necessarily correct. This reduced levels of H2AX could be due to reduced or impaired signaling and not the levels of DNA damage. If the authors want to monitor whether repair is affected, then they must do COMET assay experiments. If the comet assay is beyond the scope of this manuscript, then the wording should be changed. This is also the case in the discussion p. 18 ln. 709 when the authors refer to “reduction in DNA damage”, which is not correct.
  • In Fig. 5A the authors colony-formation data shows that siZNF84+Dox cells grow better than siNeg+Dox. Is this the case of the cells without DNA damage? Is the colony-formation capacity of the cells affected by siZNF84? Also, are these cells sensitive to DNA damage? If you use increasing concentrations of Dox, would you see a difference between the siNeg and the siZNF84? This data would be helpful in the determination of the function of this protein.
  • The authors claim the existence of a feedback loop inhibition effect regarding the expression of ZNF84, which would be repressed when the expression of p21 has achieved high levels. I don’t think that the data presented is enough to support this mechanism. Indeed, the data that supports this mechanism is contradictory on D1+16 when the expression of both genes is increased. Also previous data (Fig. 1) shows that siZNF84 causes a reduction in p21. I’d suggest tampering down this speculation as the data doesn’t completely agrees with it.